# Use of Mid-Upper Arm Circumference (MUAC) to Predict Malnutrition among Sri Lankan Schoolchildren

**DOI:** 10.3390/nu12010168

**Published:** 2020-01-07

**Authors:** Chisa Shinsugi, Deepa Gunasekara, Hidemi Takimoto

**Affiliations:** 1National Institute of Health and Nutrition, National Institutes of Biomedical Innovation, Health and Nutrition, 1-23-1 Toyama, Shinjuku-ku, Tokyo 162-8636, Japan; shinsugi@nibiohn.go.jp; 2Department of Global Health Promotion, Tokyo Medical and Dental University, 1-5-45 Yushima, Bunkyo-ku, Tokyo 113-8519, Japan; 3Department of Biochemistry and Clinical Chemistry, Faculty of Medicine, University of Kelaniya, P.O. Box 6, Thalagolla Road, Ragama 11010, Sri Lanka; dcgune@gmail.com

**Keywords:** child malnutrition, anthropometry, mid-upper arm circumference, BMI-for-age *z*-score, height-for-age *z*-score, thinness and stunting, overweight and obesity, cutoffs, schoolchildren, Sri Lanka

## Abstract

The double burden of malnutrition (under- and overnutrition) is a serious public health issue in childhood. The mid-upper arm circumference (MUAC) is a simple tool for screening nutritional status, but studies of the optimal cutoff to define malnutrition are limited. This study aimed to explore the prediction of malnutrition by MUAC in Sri Lankan schoolchildren. The participants were 538 students (202 boys, 336 girls) aged 5–10 years. Spearman’s rank correlation was calculated for MUAC and both body-mass-index-for-age z-score (BAZ) and height-for-age z-score (HAZ). Receiver operating characteristic (ROC) analysis was conducted to assess the ability of MUAC to correctly classify malnutrition, after stratifying for age and birth weight. MUAC correlated significantly with BAZ (*r* = 0.84) and HAZ (*r* = 0.35). The areas under the ROC curve for thinness, overweight, obesity, and stunting were 0.88, 0.97, 0.97, and 0.77, respectively. The optimal MUAC cutoff values for predicting thinness and stunting were 167.5 mm and 162.5 mm, respectively; the optimal cutoffs for predicting overweight and obesity were 190.5 mm and 218.0 mm, respectively. These cutoffs differed after stratification by age group and birth weight. Our results confirm MUAC to be a useful tool for monitoring growth in schoolchildren.

## 1. Introduction

Child malnutrition is a serious public health concern worldwide [1]. In urban Sri Lanka, approximately one in three primary-school children suffer the double burden of malnutrition (thinness or overweight/obesity), defined by the World Health Organization (WHO) Child Growth Standards as body mass index (BMI)-for-age z-score (BAZ) that is <−2 standard deviation (SD) and >1 SD, respectively [2]. Stunting (low height-for-age) is also recognized as a critical indicator of chronic undernutrition in assessing child growth and development, and the proportion of stunting among children aged 5–6 years old is estimated at 8.7% in Sri Lanka [3]. Early detection of and intervention against childhood malnutrition are important, given the lifelong adverse impacts of thinness, stunting, and overweight on academic performance and economic productivity [4], health-related quality of life [5], metabolic syndrome [6], and adult mortality [7].

Measurement of mid-upper arm circumference (MUAC) provides a simple and reliable tool for screening nutritional status and also enables rapid assessment of large populations in epidemiological field study. Traditionally, MUAC has served as a practical proxy measure of undernutrition and in particular, of severe acute malnutrition among infants, children under 5 years [8], and pregnant women [9]. A study of Cambodian infants under age 30 months showed the probability of acute malnutrition as defined by MUAC, varies with height-for-age z-score (HAZ) [10]. Further, repeated cohort studies have shown MUAC to be a good predictor of mortality risk in Gambian infants [11], Southeast African children and adolescents [12], and in Taiwanese older adults [13]. Despite this, there are no universally established age- and sex-specific MUAC cutoff values for identification of undernutrition in children over 5 years.

Recently, MUAC was found to be highly accurate for detecting overweight in schoolchildren in South Africa [14] and the Netherlands [15], but the cutoffs for overweight in schoolchildren established in both studies were inconsistent. There have been few studies examining MUAC cutoffs for overweight in Asian children over 5 years. Furthermore, one study in Delhi showed that birth weight affects nutritional status in schoolchildren [16]; based on this finding, a birth weight–stratified MUAC cutoff will more accurately identify vulnerable children.

In this study, we aimed to identify the MUAC cutoff values that best predict malnutrition (under- and overnutrition) in primary school children aged 5–10 years. The second aim was to obtain more accurate MUAC cutoff values for school children in Sri Lanka, where low birth weight is prevalent.

## 2. Materials and Methods

### 2.1. Study Participants

Data were collected in primary schools with grades 1–5 in the Gampaha district, an urban area of Sri Lanka, in September 2017. We categorized all primary schools into four types, according to the educational system in Sri Lanka: Type 1AB (13-year education with arts and science), Type 1C (13-year education with arts), Type 2 (11-year education), and Type 3 (5-year education). The school and class for each grade were randomly sampled from each school type. Students were selected from the index number of the attendance register using a random number table. Details of the sampling procedure have been described elsewhere [2]. In our analysis, we included 538 of 555 students who participated in the study, after excluding 17 students: 9 students whose age was not within the 5–10 years age group, and 8 students who had outlying or missing data for the main variables.

The study was approved by the Ethical Review Committee of the University of Kelaniya in Sri Lanka, the Institutional Review Board of the National Institutes of Biomedical Innovation, Health and Nutrition in Japan (No.150, June 2017), and by the Department of Education, Western Province, Sri Lanka. Informed written consent was obtained from the parents or guardians of participating children prior to participation in the study.

### 2.2. Anthropometric Assessment

The anthropometric indicators measured were height, weight, and MUAC. Height (cm) was measured to the closest 0.1 cm using a portable stadiometer (Seca, Hangzhou, China), with children standing in bare feet. Weight (kg) was measured to the closest 0.1 kg using a digital scale (Seca, Hangzhou, China), with children in light clothing. BAZ and HAZ were determined using the WHO growth curves [17]. Thinness was classified where BAZ was less than −2. Overweight was defined as BAZ > 1 whereas obesity was classified as BAZ > 2. Stunting was defined as HAZ < −2.

MUAC (cm) was measured using a colored plastic tape (TALC, Herts, UK). The trained data collectors marked the midpoint of the subject’s upper arm, located between the tips of the shoulder and elbow, then wrapped the measuring tape around the subject’s arm at the midpoint and recorded the MUAC to the nearest 0.1 cm. All measurements were performed according to the standardized protocol [18] by trained investigators on the school premises.

Additionally, birth weight data were obtained from participants’ mothers, who were asked to bring the Child Health and Development Record booklet given at birth, to refer to birth weight. Of the students who reported birth weight, 96.6% were confirmed by the booklets.

### 2.3. Statistical Analysis

For the subsequent analyses, participants were grouped into younger children (age 5–7 years) or older children (age 8–10 years). Spearman’s rank correlations were calculated for MUAC and both BAZ and HAZ to evaluate the strength and direction of any linear relationship between them. Receiver operating characteristic (ROC) curves were plotted and the area under the curve (AUC) computed to assess the sensitivity and specificity of various cutoffs for thinness, overweight, obesity, and stunting. Finally, the Youden Index was used to identify optimal cutoffs, which were stratified by age group (younger or older) and birth weight (<2500 g or ≥2500 g). The statistical analysis was performed with the Stata Statistical Software Release 15.1 macro package (StataCorp LP, College Station, TX, USA).

## 3. Results

A total of 538 schoolchildren (202 boys and 336 girls) aged 5–10 years were included in the analysis. The anthropometric characteristics of the study population, stratified by sex and age, are described in Table 1. The mean BAZ and HAZ were negative values for all the groups. Notably, the mean MUAC differed by approximately 20 mm between the two age groups. Among all groups, 5.2% of children showed stunting, and 14.9% had a history of low birth weight.

The distributions of their BMI by sex are presented in Figure 1.

### 3.1. Correlations between MUAC and BAZ and HAZ

Figure 2 shows the distribution of MUAC by BAZ and HAZ according to age group. In both age groups there was a statistically significant, strong correlation between MUAC and BAZ (*r* = 0.87 and 0.89 for the younger and older group, respectively) (*p* < 0.001). Meanwhile, there was a significant but weak correlation between MUAC and HAZ (*r* = 0.37 and 0.42 for the younger and older group, respectively) (*p* < 0.001).

### 3.2. Ability of MUAC to Correctly Identify Thinness, Overweight, and Obesity

Figure 3 illustrates the ROC curves for MUAC prediction of malnutrition (thinness, overweight, obesity, and stunting). MUAC identified overweight and obesity with high accuracy (AUC = 0.97 for both) but predicted thinness and stunting with only moderate accuracy (AUC = 0.88 and 0.77, respectively).

Table 2 shows the optimal MUAC cutoff values calculated for schoolchildren aged 5–10 years. The optimal MUAC cutoffs for the prediction of thinness, overweight, obesity, and stunting were 167.5 mm, 190.5 mm, 218.0 mm, and 162.5 mm, respectively. Notably, the MUAC cutoffs for overweight and obesity showed high sensitivity and specificity (0.85 and higher). After stratification for age and birth weight, higher optimal MUAC cutoffs were observed in the older group and in children with low birth weight.

## 4. Discussion

This study explored the use of MUAC and sought to identify cutoff values for the prediction of thinness, overweight, obesity, and stunting, in Sri Lankan schoolchildren aged 5–10 years. MUAC was found to be a highly accurate predictor of overweight and obesity and a moderately accurate predictor of thinness and stunting. Of note, different MUAC optimal cutoff values were obtained after stratifying by age group and birth weight.

With respect to the prediction of undernutrition, MUAC more accurately identified thinness than stunting. This is since MUAC measures the sum of muscle, bone, and fat in the midpoint of the arm, which are not necessarily affected by height alone. The optimal MUAC cutoff for thinness (167.5 mm) found in the Sri Lankan schoolchildren was higher than that (145 mm) previously found in African children aged 5–9 years [18]. This inconsistency may reflect racial differences but may also reflect the influence of undernutrition during late gestation, given the high rates of low birth weight (14.9%) in the present study. The higher MUAC cutoffs found in the children with low birth weight in our study further suggest that birth weight affects later nutritional status in schoolchildren. If birth weight is not taken into account, the MUAC cutoff may be overestimated for low–birth weight children.

Regarding the prediction of overnutrition, the optimal MUAC cutoffs for overweight (190.5 mm and 205.5 mm for the younger and older groups, respectively) proposed for Sri Lankan schoolchildren were consistent with the sex‒age specific MUAC cutoff values range (192–230 mm) for overweight demonstrated in Dutch children aged 5–10 years [15]. The country-specific MUAC cutoffs may also be helpful as a 12-country study showed 248 mm to be the threshold for obesity in children aged 9–11 years, with MUAC values ranging from 232 mm (South African boys) to 262 mm (British girls) [19]. In the present study, 18.8% of children were overweight according to the proposed MUAC cutoff, which is higher than the rate (12.8%) indicated by the BAZ. The higher proportion of overweight predicted by MUAC may be explained by the finer stratification in the WHO growth charts, which considers age in months—A similarly detailed MUAC reference would be needed to obtain more accurate MUAC cutoffs for schoolchildren.

Ongoing growth monitoring using this simple MUAC tool in the school setting will be helpful for all children over age 5 years in regularly checking their nutritional status. Late childhood is a critical period for healthy growth during which appropriate lifestyle habits may be learned according to nutritional status. Furthermore, in Sri Lanka, as in other Asian countries, maternal obesity has been increasing alongside the double burden of malnutrition in schoolchildren. Additional studies are also needed to determine Asian-specific cutoffs that predict malnutrition throughout life.

The results of the present study must be interpreted in light of its limitations. Firstly, the present study did not consider the influence of health status or potential morbidity/mortality risks (apart from birth weight). Moreover, as MUAC does not consider body composition, it may be a limited tool in identifying overweight and obesity in children with low skeletal muscle mass. Certainly, MUAC has often been used in extreme conditions, such as those found in refugee camps or natural disasters, where nutritional problems may be more likely; however, MUAC can be equally and easily enlisted to identify malnutrition that may occur (for example, in the context of abuse or neglect) in otherwise stable situations. Therefore, our findings reflecting the general population, as opposed to at-risk populations, are meaningful. Secondly, the study sample was drawn from a single district and was not necessarily representative of the wider Sri Lankan population, even though the participants were randomly selected in each school. Nevertheless, in the absence of international consensus on the optimal MUAC threshold for malnutrition in children over 5 years, the findings of this study provide an important contribution.

## 5. Conclusions

Our findings showed that MUAC is a good predictor of malnutrition (under- and overnutrition) in Sri Lankan schoolchildren and that MUAC cutoff values for malnutrition differ according to age group and birth weight. Easily performed, regular growth monitoring with the MUAC should be undertaken for ongoing assessment of nutritional status in schoolchildren.

## Figures and Tables

**Figure 1 nutrients-12-00168-f001:**
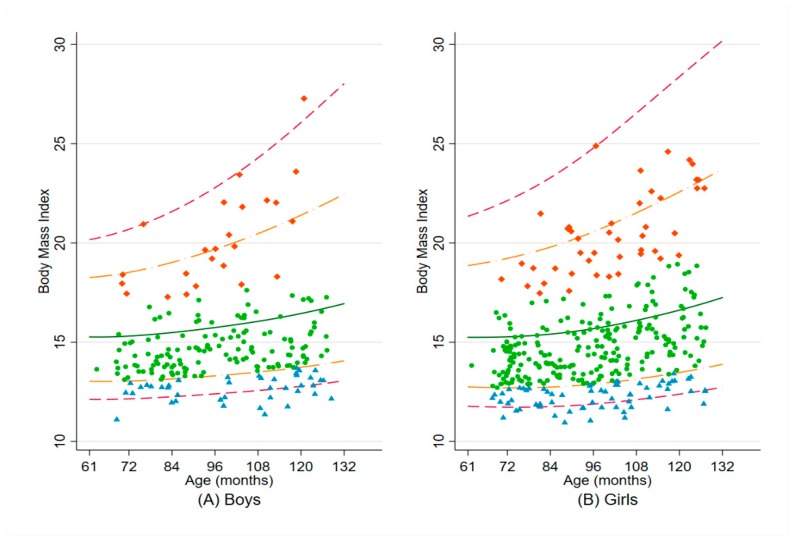
Distribution of BMI by age, from 61 to 132 months, for (**A**) Boys and (**B**) Girls. The five lines represent the median (solid dark green line) BMI for age according to the World Health Organization Child Growth Standards, and the SD from the median: ±3 SD (dashed cranberry line) and ±2 SD (long-dash and dotted orange line). Blue triangle indicates child with thinness (BAZ < −2 SD); light green dot indicates child with normal BMI for age (−2 SD ≤ BAZ ≤ +1 SD); orange-red diamond indicates child with overweight/obesity (BAZ > +1 SD). Abbreviations: BMI, body mass index; SD, standard deviation; BAZ, BMI-for-age z-score.

**Figure 2 nutrients-12-00168-f002:**
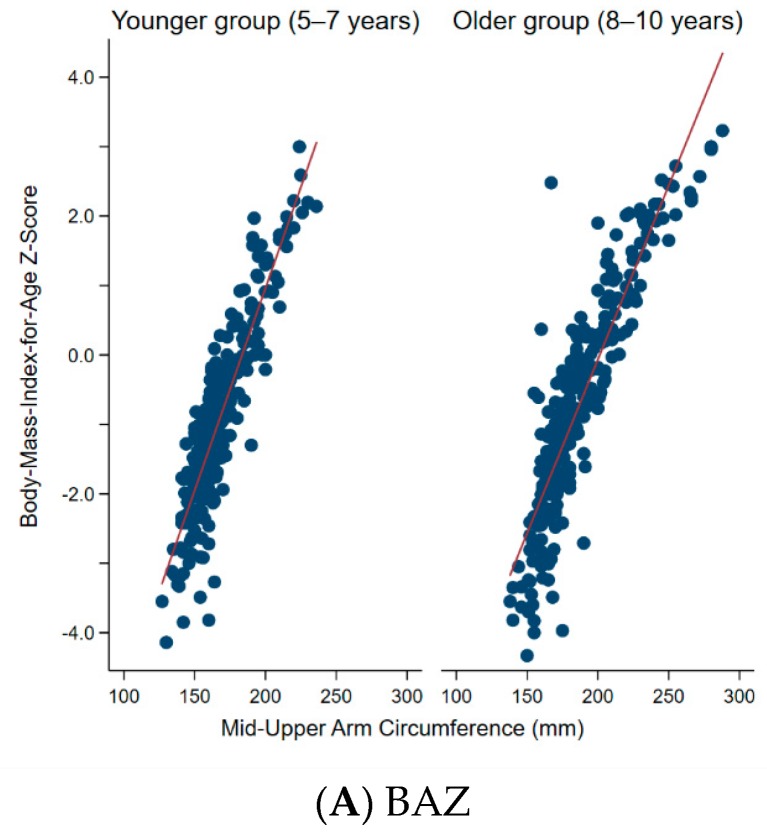
Distribution of mid-upper arm circumference by (**A**) BAZ and (**B**) HAZ, according to age group (5–7 years vs 8–10 years). (**A**) For BAZ, r = 0.87 and 0.89, for the younger and older group, respectively (*p* < 0.001). (**B**) For HAZ, r = 0.37 and 0.42, for the younger and older group, respectively (*p* < 0.001). Abbreviations: BAZ, body-mass-index-for-age z-score; HAZ, height-for-age z-score.

**Figure 3 nutrients-12-00168-f003:**
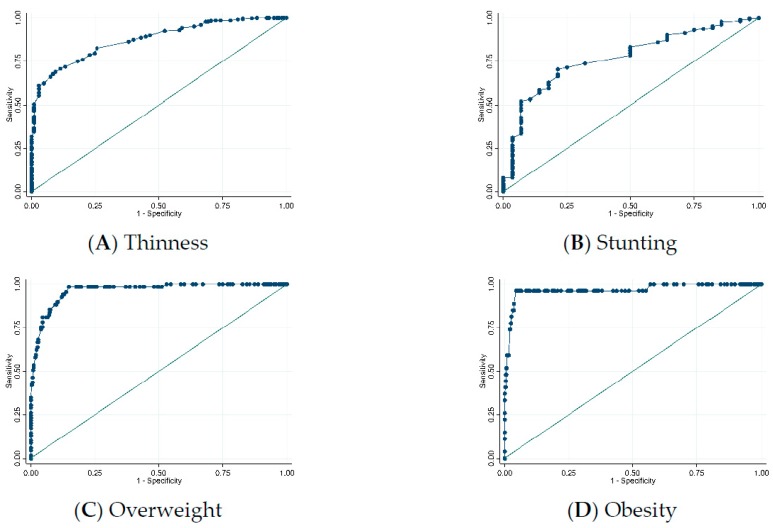
ROC curves for mid-upper arm circumference prediction of: (**A**) Thinness (BAZ < −2 SD); (**B**) Stunting (HAZ < −2 SD); (**C**) Overweight (BAZ > +1 SD); and (**D**) Obesity (BAZ > +2 SD). Abbreviations: ROC, receiver operating characteristic; BAZ, body-mass-index-for-age z-score; SD, standard deviation; HAZ, height-for-age z-score.

**Table 1 nutrients-12-00168-t001:** Anthropometric characteristics of the study population, according to sex and age group.

Variables	Total		Younger Group (5–7 Years)				Older Group (8–10 Years)			
			Boys (*n* = 94)		Girls (*n* = 165)		Boys (*n* = 108)		Girls (*n* = 171)	
	Mean	SD	Mean	SD	Mean	SD	Mean	SD	Mean	SD
Age (years)	7.6	1.5	6.2	0.7	6.2	0.7	8.9	0.8	8.8	0.8
Height (cm)	123.7	9.2	117.0	7.0	116.7	5.8	130.7	5.9	129.7	6.8
Weight (kg)	23.4	6.7	20.1	4.3	19.6	4.2	26.3	6.4	26.9	7.1
BAZ	−0.8	1.5	−0.9	1.3	−0.9	1.4	−1.0	1.7	−0.6	1.5
HAZ	−0.6	0.9	−0.7	1.1	−0.5	0.8	−0.6	0.9	−0.7	0.9
MUAC (mm)	177.8	26.6	167.9	19.4	167.8	21.3	185.0	29.5	188.3	27.2
Birth weight (%)										
<2500 g	14.9		12.8		20.0		9.3		14.6	
≥2500 g	82.9		87.2		78.8		83.3		84.2	
Missing	2.2		0.0		1.2		7.4		1.2	

Abbreviations: SD, standard deviation; BAZ, body-mass-index-for-age z-score; HAZ, height-for-age z-score; MUAC, mid-upper arm circumference.

**Table 2 nutrients-12-00168-t002:** Optimal MUAC cutoff values, with stratification by age group and birth weight, in schoolchildren aged 5–10 years (*n* = 538).

Variables	Cutoff (mm)	Sensitivity	Specificity	AUC
Thinness	167.5	0.69	0.90	0.80
Overweight	190.5	0.99	0.85	0.92
Obesity	218.0	0.96	0.96	0.96
Stunting	162.5	0.70	0.79	0.74
With Stratification by Age Group and Birth Weight:				
Thinness				
5–7 years	158.5	0.76	0.85	0.80
8–10 years	171.5	0.82	0.95	0.89
Birth weight < 2500 g	175.5	0.45	1.00	0.72
Birth weight ≥ 2500 g	166.5	0.72	0.91	0.82
Overweight				
5–7 years	190.5	1.00	0.94	0.97
8–10 years	205.5	0.95	0.92	0.93
Birth weight < 2500 g	194.0	1.00	0.97	0.99
Birth weight ≥ 2500 g	190.5	0.98	0.84	0.91

Abbreviations: MUAC, mid-upper arm circumference; *n*, number; AUC, area under the curve; BAZ, body-mass-index-for-age z-score; SD, standard deviation; HAZ, height-for-age z-score. Note: The malnutrition variables were defined as: BAZ < −2 SD (Thinness), BAZ > +1 SD (overweight), BAZ > +2 SD (obesity), and HAZ < −2 SD (stunting). There were 259 children in the younger group (5–7 years) and 279 children in the older group (8–10 years). There were 80 children with birth weight <2500 g and 446 children with birth weight ≥ 2500 g.

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
