# Peer review of "Use of Mid-Upper Arm Circumference (MUAC) to Predict Malnutrition among Sri Lankan Schoolchildren"

_nutrients, 2020, doi:10.3390/nu12010168_

Round 1

Reviewer 1 Report

The authors have conducted a cross-sectional study on 538 students (5-10 years) where they have demonstrated the usefulness of MUAC in screening for thinness, overweight, obesity and stunting.

The text is well written, clear, intelligible, with a logical succession of ideas and concepts.

The methodology used was correct. Design study, sample and recruitment of participants was appropriate.

The statistical analysis, ROC curves and cut-off points obtained of them taking into account the Youden index (highest joint sensitivity and specificity).

Results: the graphic representation was adequate and the results were clearly exposed.
Discussion: the authors have used recent references from relevant studies.

Reviewer 2 Report

This is a well-developed paper. Even though the sample is not too large to define cut-off points, the figures provided can be applied to the local population. Only small things need to be corrected or clarified. 

The objectives indicate that the purpose is to identify cut-off points for thinness and obesity. And the conclusions also refer to the same circumstances.  However, throughout the work, the prediction of chronic malnutrition is introduced, associating MUAC with growth retardation (low height for age).  Therefore, in my opinion, either the association of MUAC with height-for-age is eliminated from the analysis, or it is introduced in the objectives and conclusions. 

In figure 1(B) there is an error. On the ordinate axis (X) I think it should be Height-for age z score

 Line 35  replace  “thinness and overweight/obesity” by “ thinnes or overweight/obesity” since the same child can't have both

Reviewer 3 Report

This is an interesting paper and recognises the need for tools that can be used in the wider community to identify over and under nutrition.

Introduction: Ln 37: This sentence appears to be missing something or phrased poorly. Ln57: Statement does not align to your aims so I am unclear why you mention birth weight stratified MUAC cut-offs here. Maybe need to rephrase aim to account for this. Need to be clear that your aim is for a specific population group only as this has limited transferability to other populations without further work.

Methods: Ln64-69: This is an extremely long sentence; and repeated use of "randomly". Consider rephrasing for clarity. Ln88: Can this be guaranteed for all participants? Were they asked to produce this document to confirm the birth weight? Ln 91: was the data tested for normality prior to undertaking t-tests? Surely you would have grouped them for the t-test so this grouping is not as a result of the t-test in truth. Simply stating that they were grouped into older and younger children and presenting the t-test results in the results section would possibly be more appropriate.

Results: Figure 1: These are difficult to see and the colours are not clear (pink and red are very close).

From a reader perspective Table 1 should come before Figure 1 as the M(SD) is presented in the table but the detail is evident in the figure.

Figure 3 – you define thinness, stunting, overweight and obesity in the footnotes of the figure. This has not been mentioned anywhere in the text and should probably be explicit within the methods section. It would also be useful to know numbers (%) of children falling into each category from the total sample and by age

Ln 145: suggest “older” age group rather than higher

Discussion: Points are well made and supported. Limitation of this current study are acknowledged as are factors that may have impacted on the results when considering variation from previously published literature. One point it does not consider is body composition and the limited ability of a tool such as MUAC to identify overweight and obesity in children with low skeletal muscle mass.  
